# Dental Artifact Corruption Classifier for Head and Neck CT Images

**Prashul Singh**[1]                                                                PSINGH6@SCU.EDU

**Stephen Tambussi**[1]                                                          STAMBUSSI@SCU.EDU

**Dylan Hoover**[1]                                                                  DGHOOVER@SCU.EDU

[1] *Department of Computer Science and Engineering, Santa Clara University, USA*

**Supratik Bose**[2]                                                        SUPRATIK.BOSE@VARIAN.COM

[2] *Varian Medical Systems, Sunnyvale, USA*

**Julia A. Scott**[3]                                                                   JSCOTT1@SCU.EDU

[3] *School of Engineering, Santa Clara University, USA*

**Editors:** Under Review for MIDL 2021

## Abstract

Image artifacts emanating from dental implants can inappropriately bias the training of machine learning models for segmentation. To ameliorate the corruption of the segmentation tool, we developed and tested a dental artifact classifier to grade 2D oral cavity images as having no detectable, moderate, or severe artifactA more balanced training dataset was selected by constraining the artifact classifier to the oral cavity region. This was achieved by applying an autoencoder, which was trained only on oral cavity images, to the entire stack to determine whether an image was in the oral cavityImages with low reconstruction error were classified as oral cavity and input to a multi-class 2D convolutional neural network for the grade of an artifact. The classification was then written back into the DICOM metadata so that it may be reliably selected for subsequent usage based on artifact status. This type of approach may be applied to quality control of training datasets from uncurated sources, such as publicly available collections or de-identified patient data.

**Keywords:** Computed Tomography, Artifact detection, Autoencoders, Convolutional Neural Networks, Oral cavity.

## 1. Introduction

Analysis of medical images for radiotherapy planning is increasingly reliant upon automated, machine learning-based methods . A caveat of this methodology is the absence of inherent quality control that occurs with operators' visual inspection. The inclusion of poorly defined and artifact-ridden data sources impairs the performance of training models. This negatively impacts the development of the radiotherapy pipeline, such as for head and neck CT images used in oropharyngeal cancer treatment planning. The common occurrence of dental implants corrupts images in the oral cavity region and these images should be filtered out of training datasets used for segmentation models in the pipeline. The proposed method in this paper describes an efficient approach to selecting, grading, and classifying CT oral cavity images (Figure 1).
.

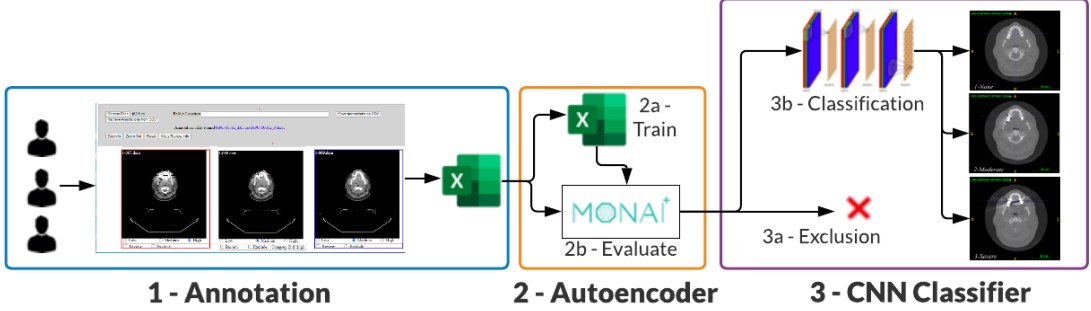

Figure 1: System architecture for dental artifact classifier.

## 2. Dataset

The models were trained on the Radiomic Biomarkers in Oropharyngeal Carcinoma (OPC-Radiomics) head and neck CT collection was accessed from The Cancer Imaging Archive (TCIA) (Cla, 2013). The dataset included 606 studies.

## 3. Dental Artifact Annotation

To meet the specific needs for visual annotation of slices by multiple raters, we built a web-based application for annotation, applying the cornerstoneTools library[1].

Each image of the stack was viewed and evaluated for artifacts. In the oral cavity region, images were labeled based on the following criteria: (1) no detectable artifact , (2) visible artifact yet surrounding structures are discernible and (3) visible artifact and surrounding structures are not discernible . The criteria are similar to other the 3-level classification schemes (Welch et al., 2020).

Four raters were trained on these classifications using The Cancer Genome Atlas Head-Neck Squamous Cell Carcinoma (TCGA-HNSC) data collection (N=2,992) The reliability was measured by Gwet's AC2 to effectively measure reliability of multiple raters for non-normal distribution of classes[2] (AC2=0.97, SE=0.006). For the model's dataset (OPC-Radiomics), each image stack was annotated by two raters. Disagreements in annotations were resolved by a third rater.

## 4. Dental Artifact Classifier

Inherent to the source of artifact constrained to the oral cavity, 95% of images in the head and neck stacks were free of dental artifact. To improve the balance of the training classifier inputs, an AE exclusively trained on oral cavity images was used to exclude non-oral cavity images based on the reconstruction error (Ng et al., 2016). The AE, built with MONAI

---

1. https://github.com/cornerstonejs/cornerstoneTools
2. Real Statistics Resource Pack, Charles Zaiontz

libraries[3], was trained on 3,475 images in batch sizes of 100 over 50 epochs resulting in a training loss of 0.0013. The reconstruction errors for non-oral cavity and oral cavity images differed by 4 to 5 fold, which separated anatomical regions with a cut-off of 2,300.

The oral cavity images used to train the AE (2,382; 91% artifact free) were input into a 2D CNN with 2 fully connected layers, 5 hidden layers, softmax activation, sparse crossentropy optimization with a batch size of 100, and run for 25 epochs. Over a 5-fold cross-validation, the mean accuracy was 95.257%+/-0.209%. The performance was above chance, yet may be improved by a more balanced and larger training set.

Due to memory constraints of the local system, the full dataset of nearly 60,000 images at a resolution of 512x512 was not able to be implemented. Further iterations of this model will first be to downsample the images to 256x256 and run through the classifier on a cloud computing service. The current code may be viewed at on a github repository[4].

## 5. Conclusion

In this multi-step approach to a dental artifact corruption classifier, we demonstrated a reliable method of multi-rater annotation, followed by an efficient filtering of the oral cavity with and without artifact, and completion of the process with a multi-class CNN to grade the artifacts. This method demonstrates a combined use of unsupervised and supervised learning to screen CT datasets for stereotyped artifacts. Implementation of this approach could be applied to other common artifacts that interfere with the performance of image processing pipelines.

## Acknowledgments

We appreciate the support of the BioInnovation and Design Lab of Santa Clara University for facilitating the collaboration with Varian Medical Systems.

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
