# OpenReview forum: "Dental artifact corruption classifier for Head and Neck CT images"
_MIDL.io/2021/Conference/Short — Submitted to MIDL 2021_

### Official Review · Reviewer_S8oN · 2021-04-21

**Confidence:** 5
**Final Rating:** 1

**Summary:**

The main contribution of this paper is an approach to identify images containing dental artifacts so that they can be removed from further training of a deep neural network for classification/segmentation purposes.

The paper uses a large number of cases from the open-source The Cancer Genome Atlas HeadNeck Squamous Cell Carcinoma (TCGA-HNSC).

**Strengths:**

Automatically identifying dental artifacts from head and neck images could be useful to improve robustness of subsequent analysis using those images either for classification or segmentation purposes.

The use of large number of datasets for analysis with cross-validation is helpful to determine generalization performance.

**Weaknesses:**

There is no methodological novelty in this work. It seems to use a standard auto-encoder.

Details of how the auto-encoder is implemented, the losses used for training, and how the classification of dental artifacts was done is not explained in the paper.



**Deanonymize Review:**

no

**Detailed Comments:**

 It's not clear what the relevance of the contribution is in clinical practice. Also, its unclear why training with images that have no dental artifacts is helpful. It helps for cases that don't contain dental artifacts. What happens for testing images with dental artifacts. Would they just be ignored? Is this clinically useful?

The methodological contribution seems to be lacking. It would be helpful to provide more details of the network architecture used, the loss functions, and how it was trained (some theory) to explain what is new here. From the description, it reads like the authors used some available technique in MONAI and trained it without any changes. This doesn't seem sufficiently novel.




**Justification Of The Rating:**

The relevance or importance of the removing dental artifact images from training is not clear. It seems like an approach that would work for a certain percentage of images but not for others. How is this clinically useful in practice?

The methodological contribution is very limited. There is nothing new proposed here. Or if there is, it could be presented better to clarify the contribution more clearly.

**Paper Type:**

validation/application paper

**Special Issue:**

no

---

### Official Review · Reviewer_ptCJ · 2021-04-27

**Confidence:** 5
**Final Rating:** 1

**Summary:**

This paper describes an approach for detecting hardening artifacts emanating from dental implants in head CT images. It consists of two main steps: An AE-based detection step for oral cavity slices, followed by an artifact severity classifier with three classes (none, mild, severe). There are no methodological advancements, but some unusual choices of techniques.

**Strengths:**

The paper is easy to understand, and the method is probably able to solve the problem (which is a little hard to say given the existing evaluation). An uncommon, but apparently adequate measure for measuring interrater agreement was chosen (Gwet's AC2).

**Weaknesses:**

* Using an AE to classify based on the reconstruction error is a strange technique. The reference given for this does not follow the same approach, so some justification would be needed. It would be more straightforward to train a classifier (as done in the reference, there based on pretrained AE features), and it may be even possible to combine this classifier with the subsequent one. The cut-off value of 2,300 was apparently chosen by hand; one may only guess that the classes are so well-separated that this may have been ok.
* The authors did not use the available dataset of 60k CT slices (from about 3k TCGA-HNSC cases), but only a small subset of 2.4k images. (The "memory constraints" should not be a problem with proper batching.)
* Even worse, the classifier was trained and cross-evaluated on the data used for training the AE.
* Finally, evaluating with "mean accuracy" is inadequate. Given that 91% of the data has the same class, 95% accuracy may be easy to reach. (A full confusion matrix may be justified?)
* When you write that "disagreements in annotations were resolved by a third rater", it is not fully specified *how*, and how that plays into Gwet's AC2 score.

**Deanonymize Review:**

yes

**Detailed Comments:**

* The CNN architecture description is incomplete (no specification of layer sizes, order, nonlinearities, etc.).
* What about early stopping? Convergence checks?
* A few formatting problems, for instance: Missing punctuation between sentences (e.g., search for "artifactA" and "cavityImages"), some spaces before punctuation (. and ,) a paragraph consisting of a lone full stop ;-) at the bottom of the first page, and the URL in footnote 4.


**Justification Of The Rating:**

The work looks useful, but as if performed by people not yet familiar with Deep Learning. Therefore, there are too many oddities that make this contribution uninteresting for the MIDL audience. In particular, as listed under weaknesses, the used subset of the dataset is too small, the evaluation is inadequate, and the AE-based classification is very odd.

**Paper Type:**

validation/application paper

**Special Issue:**

no

---

### Meta-Review · Program_Chairs · 2021-05-06

**Recommendation:** Reject
**Confidence:** 5

**Metareview:**

Both reviewers give strong reject. The authors are suggested to improve the work according to the raised comments.

---

### Decision · Program_Chairs · 2021-05-11

Reject